# Relationships between Obesity, Nutrient Supply and Primary Open Angle Glaucoma in Koreans

**DOI:** 10.3390/nu12030878

**Published:** 2020-03-24

**Authors:** Jae Yeun Lee, Joon Mo Kim, Kyoung Yong Lee, Bokyung Kim, Mi Yeon Lee, Ki Ho Park

**Affiliations:** 1Department of Ophthalmology, Sahmyook Medical Center, Seoul 02500, Korea; cielciel02@hanmail.net (J.Y.L.); myrhythm8426@naver.com (K.Y.L.); 2Department of Ophthalmology, Kangbuk Samsung Hospital, Sungkyunkwan University School of Medicine, Seoul 03181, Korea; bk415.kim@samsung.com; 3Division of Biostatistics, Department of R&D Management, Kangbuk Samsung Hospital, Sungkyunkwan University School of Medicine, Seoul 03181, Korea; my7713.lee@samsung.com; 4Department of Ophthalmology, Seoul National University Hospital, Seoul National University College of Medicine, Seoul 03080, Korea; kihopark@snu.ac.kr

**Keywords:** body mass index, diet, glaucoma, nutrition

## Abstract

To investigate the association between nutrient intake and primary open angle glaucoma (POAG) in Koreans, a population-based, cross-sectional survey, the Korean National Health and Nutrition Examination Survey, was analyzed. Glaucoma diagnosis was based on criteria established by the International Society of Geographic and Epidemiologic Ophthalmology. Multivariate regression analysis was used to assess the correlation between dietary intake and the prevalence of POAG in all enrolled subjects. In the low Body mass index(BMI) group (BMI <18.5), females with POAG had significantly lower intakes of energy, protein, fat, carbohydrate, ash, calcium, phosphorus, sodium, potassium, vitamin A, B-carotene, thiamin, riboflavin, and vitamin C than their non-glaucoma counterparts, based on a multivariate logistic regression analysis (all *p* < 0.05). In females with a medium BMI (18.5 ≤ BMI < 23), POAG showed a significant association with lower food intake, energy, protein, calcium, phosphorus, potassium, thiamin and niacin. (all *p* < 0.05). Lower protein thiamine intake in medium BMI males was related to POAG. Low dietary intake of several nutrients showed an association with glaucoma in low BMI female subjects. An insufficient intake of certain nutrients may be associated with an increased risk of glaucoma in Koreans. Further large-scale cohort studies are needed to determine how specific nutrients alter the risk of glaucoma.

## 1. Introduction

Open-angle glaucoma is a chronic progressive optic neuropathy that can cause irreversible blindness [1]. The main stress thought to be responsible for initiating glaucomatous damage is a higher intraocular pressure (IOP) than that tolerated by the optic nerve. Most current treatment methods therefore focus on reducing IOP. [2] However, glaucoma can develop or progress, even within a normal IOP range. Genetic, local, systemic, and environmental factors have also been proposed to be risk factors for glaucoma [1]. These risk factors may modify the threshold of the optic nerve to withstand glaucomatous insult. However, individuals’ optic nerve stress thresholds differ according to age, sex, race, and many other factors [3]. Glaucoma is therefore considered a multifactorial disease.

Among various risk factors, the effects of living environment and diet on glaucoma have not been widely explored. Several studies have suggested a relationship between obesity and glaucoma [4,5], while an association between low BMI and glaucoma has also been reported [5]. Several systemic risk factors have been found to be associated with glaucoma, such as metabolic syndrome [6], diabetes mellitus [7], arterial stiffness [8], and renal disease [9]. Heavy smoking [10], low estrogen level [11], and low consumption of certain fruits, vegetables, fatty fish, and walnuts [12] have been reported to be risk factors for primary open angle glaucoma (POAG) [11,12,13]. By contrast, an intake of anti-oxidative materials including flavonoids and monomeric flavanols, teas and a dietary intake of vitamins A and C have been shown to reduce the risk of POAG [13,14]. However, only weak associations were found in these studies. Moreover, race and ethnicity could affect the pathophysiologic features of POAG and POAG-related factors.

Due to the risk of glaucoma progression and blindness, there is interest in preventing glaucoma progression by means other than IOP-lowering drugs. In a recent study, we reported that glaucoma risk was negatively associated with BMI, and that high fat mass was associated with low POAG risk in Koreans [5]. Our goal in the current study was to investigate the associations between dietary nutrient intake and POAG to clarify why Koreans with low BMIs are vulnerable to glaucoma.

## 2. Materials and Methods

This survey adhered to the tenets of the Declaration of Helsinki for human research, and all participants provided written informed consent. The survey protocol was approved by the Institutional Review Board of Kangbuk Samsung Hospital.

### 2.1. Study Design and Population

The Korean National Health and Nutrition Examination Survey (KNHANES) is an ongoing, nationwide population-based, cross-sectional survey of South Koreans that is conducted by the Korea Centers for Disease Control and Prevention and the Korean Ministry of Health and Welfare [15,16]. Using a multistage, stratified, probability-clustered sampling method and weighting scheme, KNHANES provides health statistics that are representative of the civilian, non-institutionalized South Korean population. KNHANES data are deidentified and publicly available on the KNHANES website (http://knhanes.cdc.go.kr).

A total of 12,356 non-institutionalized South Koreans adults (>19 years old) were enrolled in the utilized KNHANES. Of these participants, 7746 underwent ophthalmic evaluation. Participants who showed evidence of retinal detachment, macular degeneration, or diabetic retinopathy on examination; had a history of cerebrovascular disease that could affect visual field results; or had any missing data were excluded. We also excluded individuals with a history of intraocular surgery or refractive surgery and pseudophakic and aphakic subjects, because data were not available regarding open angle status prior to cataract surgery and/or surgical history, and intraocular surgery might affect IOP or the development of glaucoma. Glaucoma patients taking glaucoma medication were not included. All 96 subjects with ocular problems and 908 subjects who had not completed the nutritional survey were excluded. A total of 6742 adults were finally enrolled for analysis.

### 2.2. Study Measurements

A general questionnaire was administered to obtain information about basic demographics, behaviors (physical activity, smoking, and alcohol consumption), and medical conditions (history of physician-diagnosed disease, current medications). All subjects were asked about medical history, alcohol consumption, and smoking status. Based on average alcohol intake per day in the month before the interview, subjects were categorized as drinkers (one or more times a week) or non-drinkers. Subjects were categorized as current smokers (more than 100 cigarettes over their lifetime and current smoking status) or non-smokers. Impaired fasting glucose was defined as fasting blood glucose >100 mg/dL but <126 mg/dL. Diabetes mellitus (DM) was defined as a fasting glucose value ≥126 mg/dL, use of oral hypoglycemic agents or insulin, or a history of DM. Prehypertension was defined as systolic blood pressure >120 mmHg but <140 mmHg or diastolic blood pressure >80 mmHg but <90 mmHg. [17] Hypertension (HT) was defined as systolic blood pressure greater than 140 mmHg, diastolic blood pressure greater than 90 mmHg, or use of antihypertensive medication for HT.

### 2.3. Nutrition Evaluation

Dietary intake was assessed using the 24-h recall method. All subjects were instructed to continue their ordinary diets before dietary evaluation. Nutrient intake was calculated on the basis of the concentration of each nutrient in a particular food using the Korean Food Composition Table, which was devised by the Korean National Rural Resources Development Institute [18]. Dietary items were categorized into 23 food groups based on classifications in the Korean Nutrient Database to simplify the interpretation of components, and the percentage of total energy intake from each food group was determined. Fiber (g/day), ash (g/day), calcium (mg/day), phosphorus (mg/day), iron (mg/day), sodium (mg/day), potassium (mg/day), β-carotene (μg/day), retinol (μg/day), thiamine (mg/day), riboflavin (mg/day), niacin (mg/day), and vitamin C (mg/day) were included in the nutrient intake data. Vitamin A (μg retinol equivalents (RE)/day) was calculated by summing retinol (μg/day) and β-carotene/6 (μg/day). Grains and their products were further divided into four subgroups: white rice, other grains, noodles and dumpling, and flour and bread.

Physical measurements of height, weight, waist circumference, and systolic and diastolic blood pressure were measured. Body mass index (BMI) was calculated as weight in kilograms divided by square of height in meters. A morning blood sample was collected after at least 12 h of fasting.

### 2.4. Ophthalmologic Examination

Ophthalmic examinations were performed by ophthalmologists. All participants underwent ocular examinations comprising visual acuity (LogMAR(Logarithm of the Minimum Angle of Resolution) Scale), slit lamp examination (Haag-streit, Inc., Bern, Switzerland), measurement of intraocular pressure (IOP) by Goldmann applanation tonometry, and fundus photography. Anterior chamber depth and anterior segment configuration were evaluated with a slit lamp. Peripheral anterior chamber depth > 1/4 peripheral corneal thickness assessed using the Van Herick method was defined as open angle. Retinal examinations of each eye were conducted based on nonmydriatic digital fundus photographs (TRCNW6S; Topcon, Tokyo, Japan). Visual field testing was performed using frequency doubling technology (FDT) perimetry (Humphrey Matrix; Carl Zeiss Meditec, Inc., Dublin, CA, USA) with the N-30-1 screening test in subjects with elevated IOP (≥22 mmHg) or a glaucomatous optic disc. POAG was diagnosed based on the International Society of Geographical and Epidemiological Ophthalmology (ISGEO) criteria and the findings of previous studies [19]. If both eyes met the criteria, data of the right eye were selected. After preliminary grading, detailed grading was performed independently by another group of glaucoma specialists, who were blinded to other data. Any discrepancy between preliminary and detailed grading was adjudicated by a third group of glaucoma specialists. Category 1 or 2 ISGEO criteria had to be met to diagnose glaucoma. For the evaluation of POAG, secondary glaucoma and childhood glaucoma were excluded in the subjects.

### 2.5. Statistical Analysis

All data were analyzed with STATA version 15.1 (StataCorp, College Station, TX, USA) to account for the complex sampling design. Strata, sampling units, and sampling weights were used to obtain point estimates and the standard error (SE) of the mean. All data were analyzed after weighting, and the SEs of mean population estimates were calculated using Taylor linearization. Participant characteristics were summarized as means and SEs for continuous variables and as frequencies and percentages for categorical variables. Demographic information and clinical parameters were compared between groups using the Pearson Chi-square test for categorical variables and the general linear model for continuous variables. All data were analyzed according to sex and BMI. Multivariate logistic regression controlling for confounding variables was used to evaluate the independent risk factors associated with OAG. Odds ratios (ORs) with 95% confidence intervals (CIs) were estimated using logistic regression models. Model 1 was adjusted for age; model 2 was adjusted for age, DM, HT, HDL, and IOP; and model 3 was adjusted for age, DM, HT, HDL, IOP, smoking, and alcohol use. A *p* value < 0.05 was considered statistically significant.

## 3. Results

Demographic information is provided in Table 1. Of 323 participants diagnosed with POAG, the average age was 50.97 ± 1.24 years, which was greater than that of subjects without POAG (42.82 ± 0.30 years). More males than females had POAG (55.77% vs. 46.39%). Systolic blood pressure and diastolic pressure were higher and HDL-C was lower in the POAG group than the group without POAG. Participants with POAG tended to have higher frequency of HT and DM than those without POAG. IOP was higher in subjects with OAG (14.39 ± 0.22 mmHg) than in those without POAG (13.96 ± 0.08 mmHg, *p* < 0.001).

Table 2 shows the overall analysis results without stratification for differences in nutrient intake between groups with and without POAG. Subjects with POAG had significantly lower protein (69.47 ± 2.65 g vs. 77.42 ± 0.80 g, *p* = 0.003) and fat (39.43 ± 2.96 g vs. 45.35 ± 0.63 g, *p* = 0.044) intake than the non-POAG group. Intake of other nutrients was not significantly different between POAG and non-POAG groups.

In multivariate logistic regression analysis stratified according to sex and BMI, non-obese females (BMI < 25) with POAG had lower intakes of food, energy, protein, fat, phosphorus, potassium, thiamin, niacin, and vitamin C than females without POAG (all *p* < 0.05; Appendix A). Protein intake was significantly lower in males with POAG (81.52 ± 3.67 g) than those without OAG (94.05 ± 1.18 g) (*p* = 0.053) (Appendix A). However, there was no significant difference in nutritional intake between obese male or female subjects (BMI ≥ 25) with POAG and those without.

The results of the multivariate logistic regression analysis stratified according to sex and BMI are shown in Table 3; Table 4. There was a significant difference in nutrient intake between females with and without POAG according to BMI (Table 3). In the low BMI group (BMI < 18.5), females with POAG had significantly lower intakes of energy, protein, fat, carbohydrate, ash, calcium, phosphorus, sodium, potassium, vitamin A, B-carotene, thiamin, riboflavin, and vitamin C than females without POAG, based on the multivariate logistic regression analysis (all *p* < 0.05). Women in the medium BMI group (18.5 ≤ BMI < 23) with POAG showed significantly lower intakes of food, energy, protein, calcium, phosphorus, potassium, thiamin and niacin than those without POAG (all *p* < 0.05) by multivariate logistic regression analysis. Among women with high BMI (23 ≤ BMI < 25), only an intake of vitamin C was significantly associated with POAG.

Table 4 shows differences in nutrient intake between males with and without POAG according to BMI. There were no significant findings in the low or high BMI groups. In men in the medium BMI group (18.5 ≤ BMI < 23), there were significant associations between protein and thiamine intake and POAG, based on the multivariate logistic regression analysis (*p* < 0.05).

## 4. Discussion

Several studies of the relationship between obesity and glaucoma have found low BMI to be associated with glaucoma [3,4]. The possible causes are low cerebrospinal fluid pressure (CSFP) or primary vascular dysregulation [20,21]. Given that the lamina cribrosa isthe main region of glaucomatous damage and is located at the junction between intraocular space and subarachnoid space, a pressure imbalance between these two regions could cause glaucomatous optic nerve damage [22]. Previous studies found that lower CSFP may act similarly to elevate IOP and increase the risk for glaucomatous damage [23,24]. BMI showed a positive association with CSFP [25,26], and therefore decreased CSFP in low BMI subjects may be a risk factor for the development of glaucoma. In this study, we hypothesized that insufficient nutrient intake in low BMI individuals would also increase the risk of developing glaucoma. We therefore analyzed the relationships between BMI, glaucoma, and nutrient intake, based on an analysis of data from a large population-based database. We found a significant relationship between intake of nutrients and POAG in the South Korean population, especially females. Non-obese subjects with POAG tended to have a lower intake of nutrients than those without POAG. In particular, in females with low BMI (BMI<18.5), those with POAG showed significantly lower intake of most nutrients, including energy, protein, fat, carbohydrate, ash, calcium, phosphorus, sodium, potassium, vitamin A, B-carotene, thiamin, riboflavin, niacin, and vitamin C than their non-glaucoma counterparts. There were many nutrients factors which show significant association with POAG in the medium and low BMI groups of females. In males, lower intakes of protein and thiamine were the only factors related to POAG, and this was only observed in males with a medium BMI. Females appeared to be more affected by nutrient intake than males, with regard to the presence or absence of glaucoma.

Glaucoma is a neurodegenerative disease caused by the apoptosis of retinal ganglion cells (RGCs) and axonal atrophy and degeneration. The prevalence of POAG shows wide regional and racial variations, and the pathogenesis of glaucoma is regarded as multifactorial, but remains poorly understood. Some previous reports reported a high prevalence of POAG with normal IOP in Asians. In the study of Iwase et al., a high prevalence of POAG was reported (3.9%), and 92% of the patients with POAG had an IOP of 21 mmHg or less, and an exceptionally high percentage of those with POAG had normal IOP [27]. In Korea, the overall prevalence of POAG based on analysis of KNHANES data was estimated to be 4.7%, and the percentage of POAG with normal IOP was 4.5% [3]. In our study, the prevalence of POAG with normal IOP among the included participants was 98.9%. Unlike POAG with high IOP, an IOP-independent mechanism contributes to pathogenesis of this disease [28]. Many factors such as genetics, age, and exogenous and endogenous factors affect the development and progression of POAG [29]. Oxidative stress is known to induce molecular damage in neuronal endothelial cells and to trigger events that lead to glaucoma [29]. Moreover, because neurons in the brain do not regenerate, the accumulation of damage may exceed the capacity of repair mechanisms, leading to glaucoma [30].

Considering that oxidative stress is involved in the mechanism of glaucoma development and recent studies suggest the possibility of effects of factors other than IOP [31,32], the effects of lifestyle or nutrient intake may contribute to the development of glaucoma. Calcium plays a role in oxidative stress, and impaired calcium regulation has been implicated in neurodegenerative diseases, including glaucoma. Wang, Singh et al. [33] reported a low risk of glaucoma in individuals with high total and dietary calcium consumption. However, Ramdas, Wolfs et al. [34] did not find any significant association between dietary calcium intake and POAG. Considering the high level of salt in the traditional Korean diet (e.g., kimchi), low levels of sodium and potassium in low BMI women may reflect low dietary intake.

Vitamins can function as antioxidants and can therefore have a potential neuroprotective effect by defending against oxidative stress in glaucoma and RGC injury [35,36]. In our study, females with POAG had lower intake of vitamin A, B-carotene, riboflavin, niacin, thiamin, and vitamin C (ascorbic acid) than those without POAG. Although there is still some debate, some studies have reported that retinol equivalents protect against POAG [34,37]. Vitamin A comprises a group of animal-derived fat-soluble retinoids, while B-carotene is a water-soluble, plant-based carotenoid that can be converted into retinol. Vitamin A is essential for immunity, reproduction, cell growth, and cell differentiation. Riboflavin is involved in metabolic processes, and riboflavin deficiency is a known cause of nutritional deficiency [38]. The protective effects of niacin against glaucoma may be due to its ability to upregulate brain-derived neurotrophic factor (BDNF), and improve vascular endothelial function by decreasing vascular oxidative stress [39,40]. Niacin is also involved as a cofactor in the conversion of arginine to citrulline, which liberates nitric oxide to regulate vascular smooth muscle tone and small vessel blood flow [41]. Jung, Kim et al. [42] also found that niacin was related to the development of glaucoma in Koreans. Thiamine is an essential element in the Krebs cycle, which generates free radicals, and thiamine deficiency is a known cause of nutritional optic neuropathy [43]. The Rotterdam eye study suggested an association between low intake of antioxidant nutrients, including retinol equivalents and thiamine, and a high risk of open angle glaucoma [34]. Vitamin C is essential for collagen synthesis and the functioning of some neurotransmitters and proteins, and it assists in the regeneration of crucial antioxidants, including glutathione and Vitamin E [44]. Ascorbic acid is found in high concentrations in the cornea and vitreous and aqueous humors [45]. It has been suggested that it plays an important role as an antioxidant in ocular tissue. Yuki, Murat et al. [46] found lower serum levels of vitamin C in Japanese patients with normal-tension glaucoma than in those with normal eyes. Wang, Singh et al. [32] suggested that the supplementary ingestion of vitamin C was associated with a reduced glaucoma risk. A meta-analysis showed that plasma vitamin C level and dietary vitamin C intake are modestly correlated (r = 0.4) [47]. In our study, in the low BMI women group, a lack of the intake of most vitamins were associated with glaucoma, however, in high BMI group, there were no associations with the dietary intake of vitamins, except vitamin C. Additionally, in the obese group, the intake of vitamins did not show any association with glaucoma.

Due to the fact that men and women have different body compositions, we stratified our data according to sex. We reasoned that obese and non-obese individuals would have different dietary intakes of nutrients, so we analyzed subjects according to BMI. In our study, the association between nutrient supply and POAG was different between males and females. Females appeared to be more affected by nutrient intake than males. In the low BMI group of females, those with POAG had lower intakes of most of nutrients than those without POAG, but this was not true for males. Some studies have reported differences in the prevalence and risk factors for OAG between males and females [3,10,48,49]. Estrogen-related factors such as the effect on IOP and their neuroprotective effects have been proposedas possible mechanisms to explain the observed sex differences in POAG prevalence [50,51]. In women, estrogen decreases greatly with age, especially at menopause. Estrogen has a neuroprotective effect and demonstrates antioxidant activity [52]. It plays an important role in regulating redox equilibrium by increasing expression of antioxidant enzymes and restoring overall antioxidant status [53]. In addition, in women, adipose tissue is a source of estrogen production through aromatization of androgens; after menopause, there is an increase in subcutaneous adipose tissue aromatase activity [54,55,56]. Nutritional support appears to be needed to address this reduction in estrogen level in women with age, and may explain why women with deficient nutrient intake are more vulnerable to glaucoma. However, the mechanism of the sex-dependent association remains unknown and requires further elucidation.

In this study, lower intakes of protein and thiamine were the only factors associated with POAG in both males and females. Kinouchi, Ishiko et al. [57] reported that an increased dietary intake of meat can reduce the risk of POAG in Japanese females. Our classification of protein included a variety of plant proteins and animal proteins, including pork, beef, and chicken. Fish consumption was assessed separately in the questionnaire. We assessed meat intake by frequency of meat consumed per week, not the amount of meat. Protein is an essential component of all types of cells. A diet higher in meat may contribute to maintenance of the central nervous system [58]. In addition, neuroprotective and regenerative factors from protein have been shown to enhance the regeneration of neurites in retinas in vitro [59].

The present study had several limitations. First, the design of the study was cross-sectional, so causal relationships could not be determined. Prospective randomized controlled trials or epidemiological cohort studies are needed to determine causal associations. Second, in this study, the visual field was examined with FDT rather than Humphrey field analysis. The FDT is known to be more sensitive to early defects. Although Humphrey visual field testing is regarded to be the gold standard test for visual field testing, FDT is a fast and reliable method of detecting glaucomatous visual field defects earlier than standard automated perimetry, which may be more appropriate for large-scale screening such as in our study [60]. Third, because this study was a cross-sectional study, diurnal IOP was not measured. The IOP of most subjects was lower than 21mmHg. So, we are not sure if our results can be applied to all glaucoma patients.

Fourth, KNHANES depends on participant interviews and involves the risk of recall bias. A serum analysis of nutrients may be needed to investigate the association between glaucoma and nutrient intake. Fifth, assessments of blood levels of nutrients and dietary intake of nutrients are not standardized, which may contribute to the lack of correlations between these factors. A strength of this study is that participants were representative of subjects in the general South Korean population.

A high IOP has been considered to be the primary factor contributing to the initiation of glaucomatous damage. However, many individuals without elevated IOP develop glaucoma; age, sex, race, and undefined individual factors all contribute to glaucoma development. Due to the fact that glaucoma is a progressive neurodegenerative disease and can induce vision loss, it is important to identify modifiable risk factors for POAG. Dietary nutrition is a modifiable factor, but its association with glaucoma has not been widely explored. Since the absorption and metabolism of food differ among individuals, it is challenging to determine associations between glaucoma pathogenesis and diet. However, large-scale studies can identify trends.

## 5. Conclusions

We found an association between nutrient intake and POAG development in Koreans. The association between deficient nutrient intake and POAG was prominent in females with low BMI. The insufficient intake of certain nutrients may be associated with an increased risk of glaucoma in Koreans. Recommendations regarding sufficient nutritional intake may help reduce the risk of glaucoma in adults with a low BMI in Korea. Further large-scale cohort studies are needed to determine how nutrient intake deficiencies contribute to glaucoma.

## Figures and Tables

**Table 1 nutrients-12-00878-t001:** Demographics of enrolled subjects. All subjects completed the health questionnaire and underwent eye examinations.

	POAG (*n* = 323)	Nonglaucoma (*n* = 6419)	*p* Value
Age (years)	50.97 (1.24)	42.82 (0.30)	<0.001 *
Sex (male, %)	55.77 (3.31)	46.39 (0.68)	0.005
Current smoker (%)	27.65 (3.21)	24.78 (0.78)	0.352 ^†^
Drinker (%)	59.6 (3.44)	59.62 (0.85)	0.997 ^†^
BMI (kg/m^2^)	23.79 (0.20)	23.71 (0.06)	0.696 *
Waist circumference (cm)	81.96 (0.61)	80.88 (0.19)	0.077 *
Systolic blood pressure (mmHg)	121.22 (1.12)	115.37 (0.29)	<0.001 *
Diastolic blood pressure (mmHg)	77.68 (0.69)	75.06 (0.21)	<0.001 *
Glucose (mg/dL)	99.59 (1.83)	94.99 (0.31)	0.013 *
Total cholesterol (mg/dL)	188.69 (3.23)	188.02 (0.64)	0.841 *
HDL-C (mg/dL)	47.97 (0.90)	49.76 (0.22)	0.048 *
LDL-C (mg/dL)	111.98 (4.71)	112.68 (0.90)	0.884 *
Triglycerides (mg/dL)	139.17 (6.16)	129.98 (1.86)	0.150 *
High glucose			<0.001 ^†^
Diabetes (%)	14.89 (2.52)	6.627 (0.39)	
Pre DM (%)	15.62 (2.23)	15.56 (0.62)	
High blood pressure			<0.001 ^†^
Hypertension (%)	33.15 (3.28)	20.36 (0.66)	
Pre-hypertension (%)	27.06 (3.13)	23.05 (0.73)	
IOP (mmHg)	14.39 (0.22)	13.96 (0.08)	0.039 *

Data are presented as mean (standard error). Participant characteristics were summarized as means and standard errors for continuous variables and as frequencies and percentages for categorical variables. * *p* values resulting from general linear model for continuous variables. ^†^
*p* values resulting from Pearson Chi-square test for categorical variables. POAG, primary open angle glaucoma; BMI, body mass index; DM, diabetes mellitus; HDL-C, high-density lipoprotein cholesterol; IOP, intraocular pressure; LDL-C, low-density lipoprotein cholesterol.

**Table 2 nutrients-12-00878-t002:** Dietary intake between primary open angle glaucoma (POAG) and non-glaucoma groups.

	POAG (*n* = 323)		Non-Glaucoma (*n* = 6419)		*p* Value
	Mean (SE)	95% CI	Mean (SE)	95% CI	
Intake of food (g)	1502.64 (63.12)	1378.49–1626.78	1577.36 (16.49)	1544.93–1609.79	0.240
Energy (kcal)	2051.89 (66.07)	1921.96–2181.82	2108.24 (17.04)	2074.72–2141.75	0.398
Protein intake (g)	69.47 (2.65)	64.26–74.68	77.42 (0.80)	75.85–78.99	0.003
Fat intake (g)	39.43 (2.96)	33.61–45.26	45.35 (0.63)	44.10–46.60	0.044
Carbohydrate intake (g)	333.74 (8.75)	316.52–350.95	328.31 (2.44)	323.51–333.12	0.543
Crude fiber intake (g)	7.67 (0.33)	7.01–8.32	7.56 (0.09)	7.37–7.75	0.753
Ash intake (g)	20.92 (0.83)	19.28–22.56	21.31 (0.22)	20.89–21.74	0.644
Calcium intake (mg)	560.07 (32.55)	496.05–624.09	534.69 (5.77)	523.35–546.03	0.443
Phosphorus intake (mg)	1204.72 (45.85)	1114.56–1294.89	1244.39 (9.61)	1225.49–1263.28	0.384
Iron intake (mg)	15.26 (0.65)	13.98–16.54	15.66 (0.26)	15.15–16.18	0.554
Sodium intake (mg)	5220.44 (234.32)	4759.63–5681.26	5286.08 (64.24)	5159.74–5412.42	0.787
Potassium intake (mg)	3026.80 (120.98)	2788.87–3264.72	3202.64 (28.65)	3146.30–3258.99	0.147
Vitamin A intake (μgRE)	829.56 (54.71)	721.97–937.14	889.09 (20.16)	849.45–928.74	0.296
B-carotene intake (μg)	4255.02 (310.81)	3643.78–4866.26	4526.76 (111.79)	4306.92–4746.60	0.400
Retinol intake (μg)	111.92 (14.75)	82.92–140.91	122.79 (5.48)	112.02–133.57	0.491
Thiamin intake (mg)	1.32 (0.06)	1.21–1.43	1.42 (0.02)	1.39–1.45	0.061
Riboflavin intake (mg)	1.26 (0.07)	1.12–1.39	1.32 (0.01)	1.29–1.34	0.370
Niacin intake (mg)	16.96 (0.77)	15.45–18.47	18.24 (0.19)	17.86–18.62	0.091
Vitamin C intake (mg)	102.95 (5.28)	92.57–113.33	113.36 (1.68)	110.06–116.66	0.060

*p* values resulting from general linear model for continuous variables. CI, confidence interval; SE, standard error.

**Table 3 nutrients-12-00878-t003:** Dietary intake between primary open angle glaucoma (POAG) and non-glaucoma group in females grouped by BMI.

Diet	BMI (kg/m^2^)								
	<18.5			18.5–23			23–25		
	Model 1 *	Model 2 ^†^	Model 3 ^‡^	Model 1 *	Model 2 ^†^	Model 3 ^‡^	Model 1 *	Model 2 ^†^	Model 3 ^‡^
Intake of food (g)	0.043	0.266	0.366	0.075	0.003	0.002	0.393	0.195	0.155
Energy (kcal)	0.007	0.016	0.006	0.284	0.062	0.044	0.473	0.313	0.295
Protein intake (g)	<0.001	0.001	0.001	0.131	0.023	0.020	0.321	0.214	0.171
Fat intake (g)	<0.001	<0.001	<0.001	0.820	0.100	0.073	0.113	0.069	0.064
Carbohydrate intake (g)	0.003	0.069	0.029	0.340	0.234	0.155	0.571	0.427	0.459
Crude fiber intake (g)	<0.001	0.084	0.056	0.631	0.706	0.594	0.159	0.070	0.068
Ash intake (g)	<0.001	0.001	0.001	0.131	0.076	0.055	0.835	0.950	0.933
Calcium intake (mg)	<0.001	0.001	0.001	0.192	0.001	<0.001	0.898	0.970	0.969
Phosphorus intake (mg)	<0.001	0.015	0.015	0.170	0.018	0.012	0.450	0.280	0.265
Iron intake (mg)	0.050	0.246	0.168	0.353	0.447	0.401	0.595	0.608	0.625
Sodium intake (mg)	0.010	0.029	0.042	0.436	0.321	0.257	0.557	0.715	0.744
Potassium intake (mg)	<0.001	0.008	0.006	0.038	0.025	0.016	0.186	0.064	0.069
Vitamin A intake (μg RE)	0.001	0.016	0.005	0.699	0.651	0.679	0.376	0.368	0.384
B-carotene intake (μg)	<0.001	0.006	0.001	0.957	0.830	0.856	0.319	0.326	0.340
Retinol intake (μg)	0.912	0.735	0.895	0.416	0.775	0.808	0.316	0.262	0.258
Thiamin intake (mg)	<0.001	<0.001	<0.001	0.350	0.045	0.027	0.206	0.105	0.099
Riboflavin intake (mg)	<0.001	<0.001	<0.001	0.826	0.352	0.285	0.646	0.498	0.482
Niacin intake (mg)	0.027	0.042	0.082	0.074	0.038	0.037	0.134	0.076	0.059
Vitamin C intake (mg)	<0.001	0.004	<0.001	0.474	0.111	0.077	0.029	0.001	0.001

OR, odds ratio. Multivariate logistic regression. * Model 1: adjusted for age ^†^ Model 2: adjusted for age, diabetes, hypertension, high-density lipoprotein cholesterol, intraocular pressure. ^‡^ Model 3: adjusted for age, diabetes, hypertension, high-density lipoprotein cholesterol, intraocular pressure, smoking, drinking alcohol.

**Table 4 nutrients-12-00878-t004:** Dietary intake between primary open angle glaucoma (POAG) and non-glaucoma group in males grouped by BMI.

Diet	BMI (kg/m^2^)								
	<18.5			18.5–23			23–25		
	Model 1 *	Model 2 ^†^	Model 3 ^‡^	Model 1 *	Model 2 ^†^	Model 3 ^‡^	Model 1 *	Model 2 ^†^	Model 3 ^‡^
Intake of food (g)	0.112	0.191	0.062	0.945	0.998	0.993	0.239	0.397	0.405
Energy (kcal)	0.087	0.143	0.061	0.277	0.117	0.117	0.844	0.981	0.962
Protein intake (g)	0.533	0.602	0.287	0.061	0.003	0.006	0.377	0.514	0.513
Fat intake (g)	0.243	0.591	0.241	0.563	0.132	0.160	0.656	0.787	0.756
Carbohydrate intake (g)	0.205	0.412	0.438	0.722	0.730	0.819	0.862	0.981	0.914
Crude fiber intake (g)	0.438	0.728	0.628	0.414	0.412	0.407	0.868	0.723	0.715
Ash intake (g)	0.464	0.880	0.652	0.534	0.253	0.287	0.434	0.431	0.416
Calcium intake (mg)	0.761	0.370	0.524	0.642	0.509	0.461	0.273	0.247	0.289
Phosphorus intake (mg)	0.755	0.988	0.768	0.415	0.175	0.251	0.877	0.922	0.860
Iron intake (mg)	0.265	0.508	0.335	0.506	0.744	0.848	0.937	0.889	0.942
Sodium intake (mg)	0.652	0.937	0.829	0.352	0.098	0.075	0.719	0.717	0.650
Potassium intake (mg)	0.631	0.927	0.875	0.987	0.887	0.966	0.187	0.317	0.293
Vitamin A intake (μg RE)	0.294	0.665	0.551	0.028	0.197	0.241	0.931	0.815	0.879
B-carotene intake (μg)	0.282	0.718	0.578	0.035	0.258	0.297	0.838	0.961	0.892
Retinol intake (μg)	0.477	0.827	0.562	0.537	0.272	0.291	0.131	0.132	0.160
Thiamin intake (mg)	0.078	0.109	0.108	0.237	0.007	0.021	0.535	0.664	0.574
Riboflavin intake (mg)	0.459	0.454	0.456	0.515	0.167	0.201	0.427	0.349	0.417
Niacin intake (mg)	0.721	0.690	0.475	0.184	0.072	0.122	0.577	0.708	0.708
Vitamin C intake (mg)	0.611	0.754	0.824	0.627	0.403	0.308	0.173	0.327	0.224

CI, confidence interval; OR, odds ratio. Multivariate logistic regression. * Model 1: adjusted for age ^†^ Model 2: adjusted for age, diabetes, hypertension, high-density lipoprotein cholesterol, intraocular pressure. ^‡^ Model 3: adjusted for age, diabetes, hypertension, high-density lipoprotein cholesterol, intraocular pressure, smoking, drinking alcohol.

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
