# Peer review of "Relationships between Obesity, Nutrient Supply and Primary Open Angle Glaucoma in Koreans"

_nutrients, 2020, doi:10.3390/nu12030878_

Round 1

Reviewer 1 Report

2-Materials and Methods

2.1 75/76 line "pseudophakic and aphakic subjects ...were excluded..." why should the state before surgery (as unknown)be an exclusion criteria?

2.2 85 line "more than 100 cigarettes over their lifetime..."  You mean, 100 cigarette a week?

2.2 88/89 line "Prehypertension ... >120...<140".    On the basis of what parameters? Bibliographic references?

2.4 117/118 line "Visual field testing ... frequency doubling technology (FDT)"the reference n.18 is quoted,

 In Foster's article (ref 18) it talks about "characteristic visual field defects" by performing the program 24-2 of the Humphrey field Analyzer 2, while different examinations and with other perimeters "should be validated".

The FDT perimeter (with the screening program N-30-1) cannot be used to identify glaucoma and normal subjects as "gold standard".

3. Results

Tab 1.  It is not specified which therapy (or the number of eye drops) patients take, nor the IOP values before the beginning of the therapy (baseline values), also to differentiate hypertensive glaucoma from normal pressure glaucoma.

It is not clear what types of glaucoma are being discussed, even considering the phrase (line 214) in the discussion "Most of the subjects in our study had POAG with normal IOP".

Tab 4.  In the top legend it's written "females" but I think you should write "males"

4. Discussion

192 line. Observations on low cerebrospinal fluid pressure and its relationship to low BMI have not been sufficiently assessed, also from considerations on the high number of normal pressure glaucomas reported in the study (but not quantified) at line 214

213/214 line. "In Korea, the prevalence of POAG with normal IOP was 2,7%" (i.e. normal tension glaucoma?), "while that in OAG cases was 77%" (Sorry I don't understand). "Most of the subject in our study had POAG with normal IOP" (It isn't clear; how many cases? Which % of total cases ?) (see also Methods)

221 line. "...lifestyle or diet may also contribute..." In later sentences the contribution of diet to the genesis of glaucoma appears controversial. So the sentence should be rewritten in a more questionable way

301 line. " propter nutritional supply is expected to reduce the risk of glaucoma in low BMI people" This statement is excessively imperative in view of all the above considerations.

Author Response

2.1 75/76 line "pseudophakic and aphakic subjects ...were excluded..." why should the state before surgery (as unknown)be an exclusion criteria?

-Because we are not certain of open angle before cataract surgery because of limited information in this survey.

Therefore we changed line 75-78 as below.

->We also excluded individuals with a history of intraocular surgery or refractive surgery and pseudophakic and aphakic subjects because data were not available regarding open angle status prior to cataract surgery and/or surgical history and intraocular surgery might affect the IOP or development of glaucoma.

2.2 85 line "more than 100 cigarettes over their lifetime..."  You mean, 100 cigarette a week?

--We meant smoker as smoked more than 100 cigarettes for all over their lifetime such as during whole their life.

Those who smoked less than 100 cigarettes in their lifetimes were considered non-smoker.

2.2 88/89 line "Prehypertension ... >120...<140".    On the basis of what parameters? Bibliographic references?

In 2003, the Seventh Report of the Joint National Committee on Prevention, Detection, Evaluation, and Treatment of High Blood Pressure (JNC7) characterized adults, not taking antihypertensive medication, with systolic blood pressure (SBP) between 120 and 139 mm Hg with diastolic blood pressure (DBP) <90 mm Hg or DBP between 80 and 89 mm Hg with SBP <140 mm Hg as having prehypertension.

.

(Chobanian AV, Bakris GL, Black HR, Cushman WC, Green LA, Izzo JL, Jones DW, Materson BJ, Oparil S, Wright JT, Roccella EJ; National Heart, Lung, and Blood Institute Joint National Committee on Prevention, Detection, Evaluation, and Treatment of High Blood Pressure; National High Blood Pressure Education Program Coordinating Committee. The seventh report of the joint national committee on prevention, detection, evaluation, and treatment of high blood pressure: the JNC 7 report. JAMA. 2003;289:2560–2572. doi: 10.1001/jama.289.19.2560.)

We added this in reference #17 at line 92.

2.4 117/118 line "Visual field testing ... frequency doubling technology (FDT)"the reference n.18 is quoted,

 In Foster's article (ref 18) it talks about "characteristic visual field defects" by performing the program 24-2 of the Humphrey field Analyzer 2, while different examinations and with other perimeters "should be validated".

The FDT perimeter (with the screening program N-30-1) cannot be used to identify glaucoma and normal subjects as "gold standard".

We agree with your opinion. FDT is not a gold standard method to evaluate visual field, but it can be used in screening and diagnosis of glaucoma in previous study as below.(1,2,3)

We added at limitation line 299-304 about the use of FDT as below.

Second, in this study, the visual field was examined with FDT rather than Humphrey field analysis. The FDT is known to more sensitive to early defects. Although Humphrey visual field testing is regarded as the gold standard test for visual field testing, FDT is a fast and reliable method of detecting glaucomatous visual field defects earlier than standard automated perimetry, which may be more appropriate for large-scale screening like as our study.

  1. Kim, K.E.; Kim, M.J.; Park, K.H.; Jeoung, J.W.; Kim, S.H.; Kim, C.Y.; Kang, S.W. Prevalence, Awareness, and Risk Factors of Primary Open-Angle Glaucoma: Korea National Health and Nutrition Examination Survey 2008-2011. Ophthalmology 2016, 123, 532-541.
  2. F.A. Medeiros, P.A. Sample, R.N. Weinreb. Frequency doubling technology perimetry abnormalities as predictors of glaucomatous visual field loss. Am J Ophthalmol, 137 (2004), pp. 863-871 
  3. Bokman CL, Pasquale LR, Parrish RK 2nd, Lee RK. Glaucoma screening in the Haitian Afro-Caribbean population of South Florida. PLoS One. 2014 Dec 30;9(12):e115942.

  1. Results

Tab 1.  It is not specified which therapy (or the number of eye drops) patients take, nor the IOP values before the beginning of the therapy (baseline values), also to differentiate hypertensive glaucoma from normal pressure glaucoma.

It is not clear what types of glaucoma are being discussed, even considering the phrase (line 214) in the discussion "Most of the subjects in our study had POAG with normal IOP".

Awareness of glaucoma of Korea from the data of KHANES was at most 8%.(1,2) Most of patients among our participants did not undergo glaucoma check-up before. Originally glaucoma patients with medication was very small and those patients was not included in our subjects.

  1. Kim KE, Kim MJ, Park KH, Jeoung JW, Kim SH, Kim CY, Kang SW; Epidemiologic Survey Committee of the Korean Ophthalmological Society. Prevalence, Awareness, and Risk Factors of Primary Open-Angle Glaucoma: Korea National Health and Nutrition Examination Survey 2008-2011. Ophthalmology. 2016 Mar;123(3):532-41
  2. Kim NR, Chin HS, Seong GJ, Kim CY; Epidemiologic Survey Committee of the Korean Ophthalmologic Society. Undiagnosed Primary Open-Angle Glaucoma in Korea: The Korean National Health and Nutrition Examination Survey 2008-2009. Ophthalmic Epidemiol. 2016 Aug;23(4):238-47

We added the sentence as below at line 78-79.

Glaucoma patients with medication was not included.

Tab 4.  In the top legend it's written "females" but I think you should write "males"

Yes, That was our mistake. We changed females to males as below

.

Table 4.Difference of diet intake between primary open angle glaucoma (POAG) and non-POAG group in females grouped by BMI.

  1. Discussion

192 line. Observations on low cerebrospinal fluid pressure and its relationship to low BMI have not been sufficiently assessed, also from considerations on the high number of normal pressure glaucomas reported in the study (but not quantified) at line 214

We explained this at line 198-203 as below.

-Given that lamina cribrosa is regarded as the main region of glaucomatous damage and it is located at the junction between intraocular space and subarachnoid space, pressure imbalance between these two regions can be cause of glaucomatous optic nerve damage.[22] Previous studies found that lower cerebrospinal fluid pressure(CSFP) may act similarly to elevated IOP and increase the risk for glaucomatous damage,[23,24].BMI showed positive association with CSFP [25,26] and therefore decreased CSFP in low BMI subjects may be a risk factor for the development of glaucoma.

213/214 line. "In Korea, the prevalence of POAG with normal IOP was 2,7%" (i.e. normal tension glaucoma?), "while that in OAG cases was 77%" (Sorry I don't understand). "Most of the subject in our study had POAG with normal IOP" (It isn't clear; how many cases? Which % of total cases ?) (see also Methods)

Sorry for making confuse.

The original sentence explained the result form Namil study of Korea,(1) not a population based study, but looks inadequate. So, in this time we explained the result from population based study which seems more adequate. We changed the sentences at line 223-225 as below.

In Korea, overall prevalence of POAG from the data of KHANES was estimated at 4.7%, and the percentage of POAG with normal IOP was 4.5%.[3] In our study, prevalence of POAG with normal IOP among the included participants was 98.9%.

  1. Kim CS, Seong GJ, Lee NH, Song KC. Prevalence of primary open-angle glaucoma in central South Korea the Namil study. Ophthalmology. 2011; 118(6):1024–30.

221 line. "...lifestyle or diet may also contribute..." In later sentences the contribution of diet to the genesis of glaucoma appears controversial. So the sentence should be rewritten in a more questionable way

We changed the sentence line 232-234 as below.

Considering that oxidative stress is involved in the mechanism of glaucoma development and recent studies suggest the possibility of effects of factors other than IOP, the effects of lifestyle or nutrient supply may contribute to the development of glaucoma.

301 line. " proper nutritional supply is expected to reduce the risk of glaucoma in low BMI people" This statement is excessively imperative in view of all the above considerations.

We changed the sentence at line 320 as below.

Therefore recommendation of sufficient nutritional supply might help to reduce the risk of glaucoma in low BMI people.

Reviewer 2 Report

Open angle glaucoma is obscurely used in the manuscript.

Introduction

The first paragraph "chronic progressive optic neuropathy" should be a description of primary open angle glaucoma. The purpose of this study is to investigate the relationships between obesity, nutrients supply and primary open angle glaucoma ? Open open angle glaucoma includes  secondary open angle glaucoma and childhood glaucoma.

Materials and methods

2.4. Ophthalmological examination

The authors should mention that secondary glaucoma and childhood glaucoma were excluded in the subjects, and include the methods to exclude them in this section.

Supplemental tables S1 and S2 are confusing. The data of table S1 and S2 are completely same, although table S1 show the data in females and table S2 show those of males. In the text (p5, line1), "non-obese females (BMI<25) with OAG ----" : The authors should mention that the tables S1 and S2 show the data from the non-obese subjects.

The title of table 4 should be revised to "----in males grouped by BMI".

Author Response

Introduction

The first paragraph "chronic progressive optic neuropathy" should be a description of primary open angle glaucoma. The purpose of this study is to investigate the relationships between obesity, nutrients supply and primary open angle glaucoma? Open angle glaucoma includes secondary open angle glaucoma and childhood glaucoma.

-We changed OAG to primary open angle glaucoma (POAG).

Materials and methods

2.4. Ophthalmological examination

The authors should mention that secondary glaucoma and childhood glaucoma were excluded in the subjects, and include the methods to exclude them in this section.

We added the sentence at method line 127-128 as below.

For evaluation of POAG, secondary glaucoma and childhood glaucoma were excluded in the subjects.

Supplemental tables S1 and S2 are confusing. The data of table S1 and S2 are completely same, although table S1 show the data in females and table S2 show those of males.

->We changed the table S1 and S2.

In the text (p5, line1), "non-obese females (BMI<25) with OAG ----" : The authors should mention that the tables S1 and S2 show the data from the non-obese subjects.

->We changed the top legend of table S1 and S2 as below.

Table S1. Difference of diet intake between primary open angle glaucoma (POAG) and non-POAG group in non-obese female group. (BMI<25)

Table S2. Difference of diet intake between primary open angle glaucoma (POAG) and non-POAG group in non-obese males. (BMI<25)

The title of table 4 should be revised to "----in males grouped by BMI".

->We changed the top legend of table 4 as below.

Table 4.Difference of diet intake between primary open angle glaucoma (POAG) and non-POAG glaucoma group in females grouped by BMI.

Round 2

Reviewer 1 Report

"In Korea, overall prevalence of POAG from the data of KHANES was estimated at 4.7%, and the percentage of POAG with normal IOP was 4.5%.[3] In our study, prevalence of POAG with normal IOP among the included participants was 98.9%".

So there are almost all normal tension glaucoma patients? without or before starting therapy? 

If so, you should be talking about normal tension glaucoma (NTG), not POAG. Several authors consider it as a specific clinical entity.

Author Response

Yes, Normal-tension glaucoma (NTG) is a form of glaucoma in which damage occurs to the optic nerve without eye pressure exceeding the normal range as 10-21 mm Hg. In this study most of subjects showed normal IOP. So basically, I agreed with your opinion.

Actually, Asian including Korean is familiar with the term normal tension glaucoma (NTG). However, some western doctors did not agree to use the term normal tension glaucoma. Glaucoma with normal IOP and glaucoma with high IOP share a certain pathophysiology, especially IOP. Whether they are same disease entity on the same spectrum is on debate. Some doctors said NTG was different disease but others did not.

And our study was cross-sectional study. IOP was measured only one time point, diurnal IOP was not measured. Therefore we cannot assure they are all NTG subjects. And, even though small portion, glaucoma with high IOP (over 21mmHg) patient was included.

We think, it is explained in detail in the method and discussion, readers will not have difficulty on understanding our study.

So, if you don’t mind, I want to make it POAG in order to avoid unnecessary arguments.

We added this on limitation section at line 318-320 as below.

Third, this study was cross-sectional study and diurnal IOP was not measured. The IOP of most subjects were lower than 21mmHg. So, we are not sure if our results can be applied to all glaucoma patients.

Reviewer 2 Report

The revised manuscript has well addressed the points that I raised in the first review. One more point should be addressed. I suggest that the sbjects should be divided into POAG and non-glaucoma groups. Therefore, POAG vs Non-glaucoma is more appropriate in Table 1 and 2. The title of Table 3 and 4 should be revised as "--- primary open angle glaucoma (POAG) and non-glaucoma group---".

Author Response

We changed as you advised.

Table 2.Difference of diet intake between primary open angle glaucoma (POAG) and non-glaucoma group.

Table 3.Difference of diet intake between primary open angle glaucoma (POAG)and non-glaucoma group in females grouped by BMI.

Table 4.Difference of diet intake between primary open angle glaucoma (POAG)and non-glaucoma group in males grouped by BMI.